# Study on the Penetration Performance of a 5.8 mm Ceramic Composite Projectile

**DOI:** 10.3390/ma14040721

**Published:** 2021-02-04

**Authors:** Kai Ren, Shunshan Feng, Zhigang Chen, Taiyong Zhao, Likui Yin, Jianping Fu

**Affiliations:** 1College of Mechatronic Engineering, North University of China, Taiyuan 030051, China; renkai1127@163.com (K.R.); ssfeng@bit.edu.cn (S.F.); 2National Defense Key Laboratory of Underground Damage Technology, North University of China, Taiyuan 030051, China; cyc@nuc.edu.cn (Z.C.); zs_991109@163.com (T.Z.); 18335178769@139.com (L.Y.); 3State Key laboratory of Explosion Science and Technology, Beijing Institute of Technology, Beijing 100081, China

**Keywords:** ceramic composite projectile, penetration, numerical simulation

## Abstract

The penetration ability of a 5.8 mm standard projectile can be improved by inserting a ZrO_2_ ceramic ball with high hardness, high temperature, and pressure resistance at its head. Thereby, a ceramic composite projectile can be formed. A depth of penetration (DOP) experiment and numerical simulation were conducted under the same condition to study the armor-piercing effectiveness of a standard projectile and ceramic composite projectile on 10 mm Rolled Homogeneous Armor (RHA) and ceramic/Kevlar composite armor, respectively. The results show that both the ceramic composite and standard projectiles penetrated the armor steel target at the same velocity (850 m/s). The perforated areas of the former (φ5 mm & φ2 mm) were 2.32 and 2.16 times larger, respectively, than those of the latter. The residual core masses of these two projectiles (φ5 mm & φ2 mm) were enhanced by 30.45% and 22.23%. Both projectiles penetrated the ceramic/Kevlar composite armor at the same velocity (750 m/s). Compared with the standard projectile, the residual core masses of the ceramic composite one (Ø5 mm & Ø2 mm) were enhanced by 12.4% and 3.6%, respectively. This paper also analyzes the penetration mechanism of the ceramic composite projectile on target plates by calculating its impact pressure. The results show that the ceramic composite projectile outperformed the standard projectile in penetration tests. The research results are instructive in promoting the application of the ZrO_2_ ceramic composite in an armor-piercing projectile design.

## 1. Introduction

In modern wars, ceramic material has been primarily used in armor design owing to its outstanding performance under moderate and high-speed shock. Besides, there has been extensive research into the resistance of ceramic material under projectile impact [1,2,3]. During the penetration process, the armor-piercing projectile is prone to severe erosion, performance degradation, breakage, or deflection, which would cause its kinetic energy to decrease afterward and undermine its penetration ability. However, little can be done to improve its performance because it is typically made up of metal and alloy. When using tungsten alloy material as the core material to penetrate the ceramic target plate, it produces high stress due to the collision at the moment of contacting the target plate, resulting in core head deformation and the thickening of the warhead. These increase the resistance of penetration, thus affecting the final damage effect. Depleted uranium material as the core can self-sharpen with respect to penetration, but it can cause more severe pollution. Therefore, it is important to find alternatives that can improve the penetration capability of the projectile. Therefore, a ceramic damaging component, noted for its high performance and low cost, has drawn wide attention from researchers. Most of the chemical bonds of ceramic materials are covalent and ionic bonds, which are strong and highly directional. The hardness of ceramic materials is tens or even hundreds of times higher than that of ordinary metals. Moreover, the toughened ceramic materials not only have high hardness, melting point, compressive strength, wear and corrosion resistance, and toughness, but also have lower costs.

Research on the military application of ceramic material has focused on the improvement of its protective capability. In contrast, few have studied how the penetration performance of armor-piercing projectile will be affected when ceramic material is applied. For example, based on numerical simulation, Nechitailo [4,5] adopted pre-stress to prevent the ceramic projectile from breaking during the penetration process. They also designed a penetration experiment where concrete targets were penetrated by steel projectiles of the same diameter, mass, and length with diamonds of different masses inserted on their points. 

The results showed that the penetration depth of the projectiles is proportional to the diamond mass and hit velocity, indicating that, at high penetration speed, diamond-inserted projectiles penetrate deeper than steel ones. Li [6] preliminarily analyzed the impact of two kinds of cylinders, namely, ceramic and alloy steel, on ceramic/composite material target plates. They found that, compared with the steel component, the ceramic one caused much more severe damage to the bulletproof ceramic plate. Wang [7] conducted using a numerical simulation and experiments to study the impact of a Ø7 mm toughened Al_2_O_3_ ceramic ball on ceramic composite armor. Rui [8] studied the cratering effect of ceramic composite bullets on concrete. Huang [9] studied the velocity attenuation law of spherical ceramic breakers. Yi [10] and Li [11] et al. designed a low-intrusion ceramic bullet and studied the low-intrusion bullet’s ability to penetrate aircraft bulkheads and portholes.

Based on previous studies, this study applied a high-hardness ZrO_2_ ceramic to 5.8 mm standard projectiles to examine its penetration mechanism with respect to Rolled Homogeneous Armor (RHA) and ceramic/Kevlar composite armor through a DOP experiment.

## 2. Experiment Preparation

### 2.1. Material Preparation and Property Analysis

Ceramic material possesses unique strengths, but its performance is affected by its brittleness, which is primarily because of two factors. Firstly, when being sintered at 800–1000 °C, ceramics have high porosity [12]. Thus, pores of various sizes would be formed on the fracture surface of the ceramic, and particles are comparably larger. If the temperature range is 1200–1400 °C, the particle structure would be finer. The second factor is high energy bonds in the microstructure of ceramics. Ceramic material is primarily constituted of ionic bonds and contains a small number of covalent bonds and metallic bonds, making this material rigid, brittle, and easy to crack. Lowering the imperfection susceptibility of ceramics by compounding them with reinforcements of smaller defect sizes (such as ZrO_2_ or SiN) can enhance their strength because the method can significantly improve the fracture toughness and the critical stress for crack propagation [13,14,15,16].

A ZrO_2_ ceramic ball applied as the point of the ceramic composite projectile is produced by gel casting [17,18,19], where 3% (mole fraction) Al_2_O_3_ ceramic powder is added into the matrix material of ZrO_2_ to increase its density to 5.9 g/cm^3^. Firstly, the organic monomer (14.5% acrylamide), cross-linking agent (0.5% methylene bisacrylamide), and deionized water were mixed into the pre-mixed solution. Then, a certain amount of ZrO_2_ ceramic powder and dispersing agent (ammonium citrate) were added for 24 h of ball milling to obtain a ceramic slurry with a solid loading of 44%. Then, a catalyst (0.16 mL/g tetramethylethylenediamine) and initiator (0.16 mL/g ammonium persulfate) were added and mixed evenly before being injected into the mold. The cured ZrO_2_ body was processed according to the designed size. Finally, the blanks are sintered at 1520 °C to obtain two ZrO_2_ ceramic spheres, with one having a diameter of 5 mm and a mass of 0.38 g and another having a diameter of 2 mm and a mass of 0.03 g.

By analyzing the scanning electron microscopy (SEM, CARL ZEISS, Heidenheim, Germany) results, it can be seen from the microtopography (Figure 1a) that the Zirconia Toughened Ceramics is more even, and no defects such as pores or cracks are found. The crystalline gains are regular in shape and even in size, and the grain boundary is comparably narrow, ranging from 2 to 3 μm. By contrast, in the Al_2_O_3_ ceramic, the grain diameter ranged between 1–15 μm and there were more randomly distributed pores in the sample with diameters ranging between 1 μm and 15 μm. As Figure 1b shows, the size of the pores are similar to that of the grains [20,21]. Tests showed that the toughened ZrO_2_ ceramic has a relative density of 91.2%, bending strength of 850 MPa, and fracture toughness Kic of 9.35 MPa/m^2^.

The fracture mode, which can be classified into intergranular fracture and transgranular fracture, is closely related to its strength. According to Figure 2b, the A1_2_O_3_ ceramic presents distinct characteristics of the intergranular fracture of Al_2_O_3_ granules, showing a “crystal sugar-like” fracture caused by crack propagation along the A1_2_O_3_ grain boundary [22,23]. Figure 2a shows that the fracture modes of the ZrO_2_-toughened A1_2_O_3_ ceramic include intergranular fracture and transgranular fracture. As the fracture propagation under transgranular fracture mode consumes more energy than that under the intergranular one, its presence will enhance the connection strength between grains [12].

### 2.2. Depth-of-Penetration Experiment

As shown in Figure 3, the test used a 5.8 mm projectile of three different structures. They are (from left to right) the standard projectile core, standard projectile core with a Ø2 mm ZrO_2_ ceramic ball, and standard projectile core with a Ø5 mm ZrO_2_ ceramic ball, respectively. All projectiles have the same total mass of 4.5 g. The core diameters of the three structures are 5 mm, and the core lengths are 15 mm, 15 mm, and 14.2 mm, respectively. Given that the copper backing armor and lead jacket have little effect on the projectile penetration capability, in the tests and simulations, the three structures are simplified to the extent that only the cores and ceramic balls are included.

The applied targets were 10 mm Rolled Homogeneous Armor (RHA) and ceramic/Kevlar composite armor. The ceramic/Kevlar composite armor has two layers of Kevlar on its front, Al_2_O_3_ ceramic with 10 mm thick and 50 mm × 50 mm side length in the middle, and 10 layers of Kevlar on its back. The Al_2_O_3_ ceramics were glued together with Kevlar on both sides. This experiment adopted a 12.7 mm smoothbore ballistic gun and a laser velocimeter independently developed by North University of China (Taiyuan, China). The test is shown in Figure 4. The three projectiles were loaded with nylon cartridges and were fired from a 1.27 mm ballistic gun. The muzzle velocity of the projectile was controlled by adjusting the charge of the cartridge.

## 3. Simulation Calculation

### 3.1. Simulation Model

TrueGrid software (v3.13) was applied to construct the finite element model of the ceramic composite and standard projectile (Ø5 mm) (as shown in Figure 5) to compare and analyze their penetration ability against the RHA and ceramic/Kevlar composite armor target. AUTODYN software (19.0) was used to calculate the result. Smoothed Particle Hydrodynamics (SPH) was adopted for ZrO_2_ ceramic balls and ceramic target plates. The Lagrangian algorithm was used for the tungsten core, RHA target plate, Kevlar panel, and back panel. The radius of influence of SPH particles and the size of the Lagrange grid are both 0.1 mm. The simulation used the CONTACT_ERODING_SURFACE_ TO_SURFAC algorithm to solve the grid distortion that may occur during the action of the projectile and the target plate, and the CONTACT_AUTOMATIC_SURFACE_TO_SURFACE algorithm was used between the ZrO_2_ ceramic and the tungsten core. The pressure outflow boundary condition was exerted on the model boundary, which was equivalent to the circumferential stress constraint effect.

### 3.2. Parameters of Materials

In the finite element calculation, the JH-2 material model was adopted for ZrO_2_ ceramics and ceramic target materials. The Johnson–Cook material model was applied to projectile core material (tungsten alloy) and RHA material. The Gnuieisen state equation was employed to describe the dynamic response behavior. The Puff state equation and Von Mises strength model were used for Kevlar. The parameters of the ZrO_2_ ceramic constitutive model can be seen in Table 1. Other parameters are shown in the references [24,25,26].

## 4. Results and Analysis

### 4.1. The Penetration of Armored Steel Target

#### 4.1.1. Test Results

Projectiles with three structures were used to penetrate a 10 mm homogeneous armored steel target at an initial speed of 850 m/s to determine their penetration effect on armored steel targets at high speed. The test data are shown in Table 2. Figure 6 shows the damage to the front and back target plate.

Table 2 shows the damage effect data of 1^#^–9^#^ projectiles for the armored steel target. Through comparison, it is found that the penetration aperture of the ceramic composite projectile on the armored steel target is larger than that of the standard one. Furthermore, the larger the ceramic balls, the larger the aperture and flange. This is because during the cratering process, the ZrO_2_ ceramic balls were crushed under extremely high pressure, which enlarged the contact area between the projectile core and target plate. Therefore, the aperture formed on the target plate is larger than the standard projectile core. As indicated in Figure 6, after the aperture of the target plate is measured, it is found that the average penetration areas of the ceramic composite projectiles (Ø5 mm & Ø2 mm) are 2.46 times and 2.14 times, respectively, that of the standard one. As shown in Figure 7, the average residual masses of projectile cores recycled from the ceramic composite projectile (Ø5 mm), ceramic composite projectile (Ø2 mm), and standard projectile are 3.61 g, 3.17 g, and 2.19 g respectively, accounting for 80.15%, 70.52%, and 48.74% of the total. The residual mass of the ceramic composite projectile core is remarkably larger than that of the standard one.

#### 4.1.2. Simulation Analysis

Figure 8 displays the comparison of the standard projectile and ceramic composite projectile (Ø5 mm) when penetrating to the half-depth of the armored steel target at an initial speed of 850 m/s. As shown in Figure 8, as two various projectiles penetrate to the half-depth of an armored steel target, the shape of the ceramic composite projectile (Ø5 mm) core barely changed. In comparison, the standard core head is severely flattened and eroded. The residual length of the ceramic composite projectile (Ø5 mm) core is prominently longer than that of the standard one.

Figure 9 and Figure 10 show the mass-time curve and the velocity/acceleration-time curve of two projectiles in different structures. It can be seen from Figure 9 and Figure 10 that, in terms of the residual mass and velocity, the ceramic composite projectile (Ø5 mm) core outperforms standard one during the penetration. The residual masses of the two projectile cores are 3.62 g and 2.28 g, respectively. The remaining velocities after the penetration are 359.3 m/s and 232.7 m/s, respectively, similar to the test results. During the penetration process, the ceramic warhead of the ceramic composite projectile (Ø5 mm) acted on the target plate in advance, thus effectively protecting the projectile core. Therefore, its loss in mass and velocity is less than those of the standard projectile. When t = 18 μs, the core mass of standard projectile decreases sharply. Meanwhile, due to the mushroom erosion of the standard projectile core, the target plate increases its resistance to the core and aggregates the abrasion. By comparing the accelerated velocity-time curve, it can be seen that acceleration of the ceramic composite projectile (Ø5 mm) core is lower than that of the standard one during the penetration, which indicates that the load of the former during the penetration is lower.

Figure 11 shows the simulation damage effect of the two projectiles on the arm steel target. By comparing the apertures of two varied projectiles, we found that the ceramic composite projectile is larger than the standard projectile in terms of hole entrance, hole exit, and maximum hole diameter.

As the test and simulation results show (Table 3), the ceramic composite projectile (Ø5 mm) is superior to the standard one in terms of aperture, residual mass, and residual velocity because it effectively protected the core during the penetration. 

### 4.2. The Penetration of the Ceramic/Kevlar Composite Armor Target

#### 4.2.1. Test Results

Table 4 shows the test data for the three projectiles with different configurations penetrating the ceramic/Kevlar composite armor at 750 m/s.

According to the test results, all projectiles penetrated the ceramic layer of the ceramic/Kevlar composite armor target, thus leading to fragments and bulges of the Kevlar fiber layer on the back. The damage effect of the ceramic composite projectile (Ø5 mm) on the ceramic/Kevlar composite armor is greater than that of the ceramic composite projectile (Ø2 mm) and standard projectile. By comparing the damage of the target plate, we found that the ceramic composite projectile exerted more severe damage on the target due to its strong hardness and the extreme impact force produced from the instant impact. Figure 12 shows the damage on the Kevlar surface, which proves that the ceramic composite projectile (Ø5 mm & Ø2 mm) suffers more severe damage than the standard one. According to the broken situation of ceramics in Figure 13, the aperture of the ceramic cone generated by the penetration of the ceramic composite projectile (Ø5 mm & Ø2 mm) to target plates is prominently larger than that of the standard projectile. As shown in Figure 14, the average residual masses of recycled project cores from the ceramic composite projectile (Ø5 mm), ceramic composite projectile (Ø2 mm), and standard projectile are 1.63 g, 1.27 g, and 1.09 g respectively, accounting for 36.1%, 28.2%, and 24.2% of the total.

#### 4.2.2. Analysis of the Simulation Results

In the simulation analysis, both the standard projectile and ceramic composite projectile (Ø5 mm) penetrated the ceramic/Kevlar composite armor at an initial velocity of 750 m/s. Figure 15 shows the penetration process of the latter, which started when t = 2 μs and its ZrO_2_ ceramic head with high rigidity acted on the upper layer of the Kevlar. Meanwhile, the head was squashed under high pressure. Due to the small contact area between the projectile head and Kevlar, the surface stress concentrated on the upper layer and penetrated it. When t = 15 μs, a ceramic cone (61°) was formed in the ceramic layer under the impact of the ceramic composite projectile (Ø5 mm). When t = 55 μs, the velocity of this projectile was reduced to 0. At this time, the projectile core penetrated through the ceramic layer, and the Kevlar board showed severe erosion.

Figure 16 and Figure 17 are the mass–time and velocity/acceleration–time curves of the two projectile cores. The simulation results showed that the standard projectile and ceramic composite projectile (Ø5 mm) failed to penetrate through the ceramic/Kevlar composite armor at the velocity of 750 m/s. The core masses residuals were 1.061 g and 1.724 g, respectively, accounting for 23.6% and 36.1% of their total masses. The residual mass ratio of the ceramic composite projectile (Ø5 mm) was 14.7% higher than that of the standard projectile. The results of the simulation and experiment were consistent. As shown in Figure 16 and Figure 17, at the beginning of penetration, the mass and velocity of standard projectile core decreased faster than those of 5 mm composite projectile core. This demonstrated that in this process, the latter bore a smaller load, which is because the ceramic head acted on the target first when the ceramic composite projectile (Ø5 mm) contacted the target. Thus, the metal core was protected. 

According to the experimental and simulation results, the ceramic head of high rigidity protected the metal core to a certain extent. As a result, the residual core mass of the ceramic composite projectile (Ø5 mm) was larger than that of the standard projectile after the penetration. Besides, the penetration effect of the former was slightly better. Therefore, the structure of the ceramic head shall be optimized further so that the composite projectile can penetrate through the ceramic/Kevlar composite armor.

## 5. Impact Pressure Calculation Model for the Penetration of the Ceramic Composite Projectile into the Target 

### 5.1. Basic Relational Expression

After object II hits object I at velocity $u_{0}$, two shock waves would be generated in the two objects. The shock wave velocities are $D_{1}$ and $D_{2}$, respectively, as shown in Figure 18. Thus, the mass and momentum conservation conditions across $D_{1}$ and $D_{2}$ are as follows:

Right wave:(1)$$\rho_{1}(D_{1}-u_{1})=\rho_{10}(D_{1}-u_{10})$$
(2)$$P_{1}+\rho_{1}{\left(D_{1}-u_{1}\right)}^{2}=P_{10}+\rho_{10}{\left(D_{1}-u_{10}\right)}^{2}$$

Left wave:(3)$$\rho_{2}(D_{2}+u_{2})=\rho_{20}(D_{2}+u_{20})$$
(4)$$P_{2}+\rho_{2}{\left(D_{2}+u_{2}\right)}^{2}=P_{20}+\rho_{20}{\left(D_{2}+u_{20}\right)}^{2}$$

It is assumed that the absolute velocity $D_{2}$ is greater than 0.

The equation group composed of Equations (1)–(4) contains 14 quantities, including eight unknown ones ($D_{2}$, $u_{2}$, $\rho_{2}$, $P_{2}$, $D_{1}$, $u_{1}$, $\rho_{1}$ and $P_{1}$) and (6) known ones ($u_{10}=0$, $u_{20}=u_{0}$ and $P_{10}=P_{20}=0$). There into, $\rho_{10}$ and $\rho_{20}$ refer to the density of these two objects.

The boundary conditions are:(5)$$P_{1}=P_{2}$$
(6)u=1u2

The materials in the Hugoniont state [27,28,29,30] are shown as: (7)$$P_{1}=P_{1H}(\rho_{1H})\;\;\rho_{1H}=\rho_{1}$$
(8)$$P_{2}=P_{2H}(\rho_{2H})\;\;\rho_{2H}=\rho_{2}$$

The equation group composed of Equations (1)–(8) is closed and can be solved after the above initial conditions are given.

### 5.2. Shock Wave and Particle Velocity

Many experiments have shown that the elastoplastic wave will become a single strong shock wave under high-velocity impact. The particle and shock wave velocities can satisfy a linear relation; therefore:(9)D=a+bu

The above velocities of the left wave and mass point do not satisfy the linear relation. Therefore, only the relation between pressure and density on the shock front can be deduced.

If object I is static before the collision, which means $u_{10}=0$, it can be obtained from Equation (1) that:(10)u=1(1−ρ10ρ1)D1

If $\eta_{1}=1-\frac{\rho_{10}}{\rho_{1}}$, u=1η1D1 will be obtained from the above equation. Thus, η1=u1D1.

It can be obtained from the linear relational Equation (9) that:(11)$$D_{1}=a_{1}+b_{1}u_{1}=a_{1}+b_{1}\eta_{1}D_{1}\;\;\mathrm{or}\;\;D_{1}=\frac{a_{1}}{1-b_{1}\eta_{1}}$$
(12)$$u_{1}=\frac{\eta_{1}a_{1}}{1-b_{1}\eta_{1}}$$

In this way, Equations (11) and (12) can associate the changes of particle velocity, shock wave velocity, and density.

By substituting Equations (11) and (12) into (2), the following equations can be obtained:(13)$$P_{1}=\rho_{10}D_{1}u_{1}$$
(14)$$P_{1}=\rho_{10}\frac{\eta_{1}a_{1}^{2}}{{\left(1-b_{1}\eta_{1}\right)}^{2}}$$

The equations represent the relation between pressure and density, which is known as the Hugoniot relation of shock-loaded matter. It indicates that the shock wave pressure is related to different densities, but has no connection with the direction of the shock motion. Therefore, there is a corresponding relational expression for the left wave:(15)$$P_{2}=\rho_{20}\frac{\eta_{2}a_{2}^{2}}{{\left(1-b_{2}\eta_{2}\right)}^{2}}$$

In Equation (15):(16)$$\eta_{2}=1-\frac{\rho_{20}}{\rho_{2}}$$

According to Equations (3) and (4), the relation of $u_{2}$ and $\eta_{2}$ is shown as:(17)$$u_{2}=u_{20}+\frac{\eta_{2}a_{2}}{1-b_{2}\eta_{2}}$$

Further, the interfacial pressure is linearly dependent on the velocity. On this basis, the nonlinear equations to solve $\eta_{1}$ and $\eta_{2}$ can be obtained by combining Equations (12), (14), (15), and (17), as shown below:(18)$$\left\{\begin{array}{c} \rho_{10}\frac{\eta_{1}a_{1}^{2}}{{\left(1-b_{1}\eta_{1}\right)}^{2}}=\rho_{20}\frac{\eta_{2}a_{2}^{2}}{{\left(1-b_{2}\eta_{2}\right)}^{2}}\\ u_{20}+\frac{\eta_{2}a_{2}}{1-b_{2}\eta_{2}}=\frac{\eta_{1}a_{1}}{1-b_{1}\eta_{1}} \end{array}\right.$$

A binary nonlinear equation like this cannot figure out the relation of $\eta_{1}$, $\eta_{2}$ with $u_{20},a_{1},b_{1},a_{2},b_{2},\rho_{10},\rho_{20}$ directly. However, $\eta_{1}$ and $\eta_{2}$ can be solved firstly through the numerical method.

The shock wave pressure ($P_{1}$,$P_{2}$) under different penetration velocities can be solved by substituting $\eta_{1}$ and $\eta_{2}$ into Equation (14) and (15).

### 5.3. Calculation Results

Ceramic-composite projectile and standard projectile impacted the RHA and ceramic/Kevlar composite armor at the same velocity. The Hugoniot parameters of materials are shown in Table 5 [31].

Material parameters were substituted into Equations (18) to first calculate $\eta_{1}$ and $\eta_{2}$ through Lingo and then solve the instantaneous impact pressure when the projectile penetrates the target. At the velocity of 850 m/s, the immediate impact pressure of the ceramic composite projectile on the armor steel target was 1.83 GPa higher than that of the standard projectile. At the velocity of 750 m/s, the instantaneous impact pressure of the former on the ceramic/Kevlar composite armor was 1.32 Armor higher than that of the latter. At the velocity of 850 m/s, the instantaneous impact pressures of the ceramic composite projectile and the standard projectile on the armored steel target were 33.39 Armor and 31.54 Armor, respectively. At the velocity of 750 m/s, the instantaneous impact pressures of the ceramic composite projectile and the standard projectile on the ceramic/Kevlar composite armor were 14.29 Armor and 12.97 Armor, respectively.

## 6. Conclusions

This paper combined experiments, numerical simulations, theoretical analyses, and analysis models to analyze the vertical penetration of a ceramic composite projectile (Ø5.8 mm) into an armor steel target (10 mm) and a ceramic/Kevlar composite. The results show that:(1)The ceramic composite projectile and standard projectile penetrated the armor steel target at the same velocity (850 m/s). The perforated areas of the ceramic composite projectiles (Ø5 mm & Ø2 mm) were 2.32 and 2.16 times larger, respectively, than that of standard projectile. The residual core masses of the two projectiles (Ø5 mm & Ø2 mm) were enhanced by 30.45% and 22.23%, respectively. The result showed that when penetrating RHA, the performance of the ceramic composite projectile was much better than that of the standard projectile.(2)The ceramic composite projectile and standard projectile penetrated the ceramic/Kevlar composite armor at the same velocity (750 m/s). The former had a better effect on the Kevlar board and ceramic board. Besides, compared with standard projectile, the residual core masses of the ceramic composite projectiles (Ø5 mm & Ø2 mm) were enhanced by 12.4% and 3.6%, respectively. However, the three projectiles failed to penetrate through the ceramic/ Kevlar composite target. Therefore, the structure of the ceramic head shall be optimized.(3)Compared with the Ø2 mm ceramic composite projectile, the Ø5 mm ceramic projectile showed a much better penetration effect on the RHA and ceramic/Kevlar composite armor, which indicates that the larger the ceramic head, the better the penetration effect.(4)While penetrating the target, the ceramic composite projectile generated a larger instantaneous impact pressure than the standard projectile. The damage effect of the former was better than that of the latter.(5)Compared with the standard projectile, the ceramic composite projectile had a better penetration effect. Therefore, ZrO_2_ ceramic materials have a promising application prospect in projectile design in the field of armor-piercing penetration.

## Figures and Tables

**Figure 1 materials-14-00721-f001:**
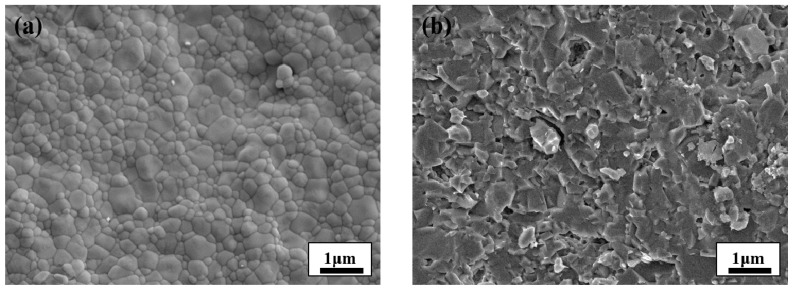
SEM electronic scanning images: (**a**) ZrO_2_ Ceramics; (**b**) Al_2_O_3_ Ceramics.

**Figure 2 materials-14-00721-f002:**
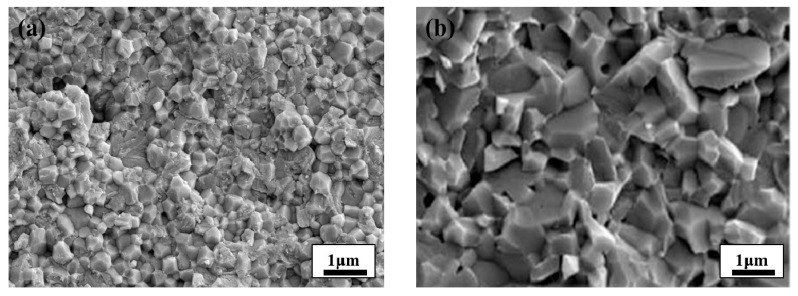
Fracture Morphology Images: (**a**) ZrO_2_ Ceramics; (**b**) Al_2_O_3_ Ceramics.

**Figure 3 materials-14-00721-f003:**
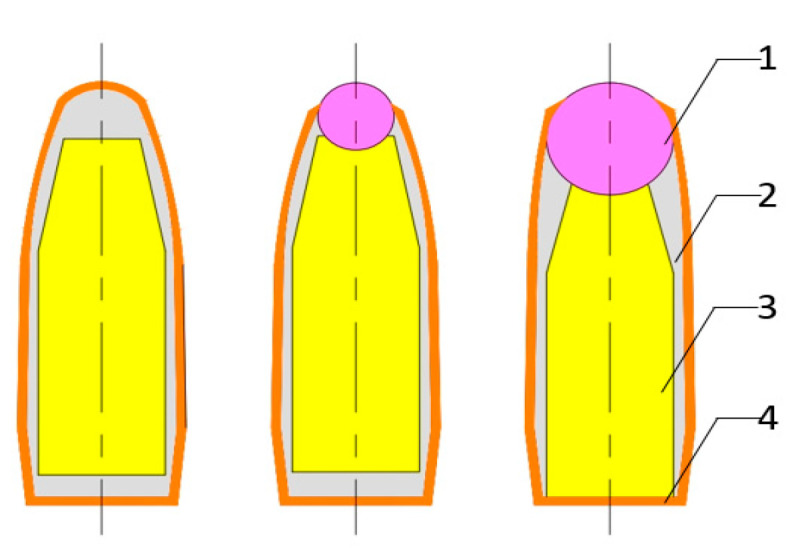
Three projectiles with different structures. 1 ZrO_2_ ceramic ball, 2 Lead case, 3 Tungsten core, 4 copper jacket.

**Figure 4 materials-14-00721-f004:**
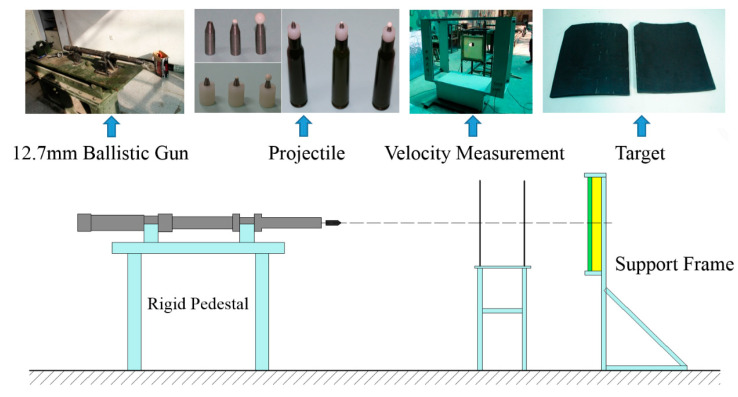
Schematic of the experimental setup.

**Figure 5 materials-14-00721-f005:**
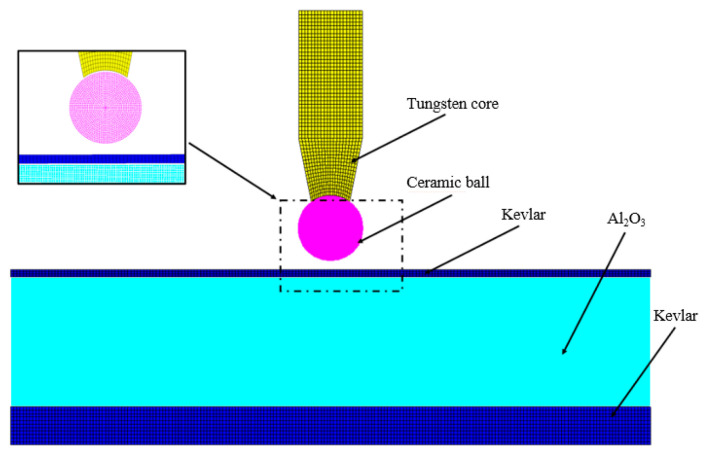
Simulation model.

**Figure 6 materials-14-00721-f006:**
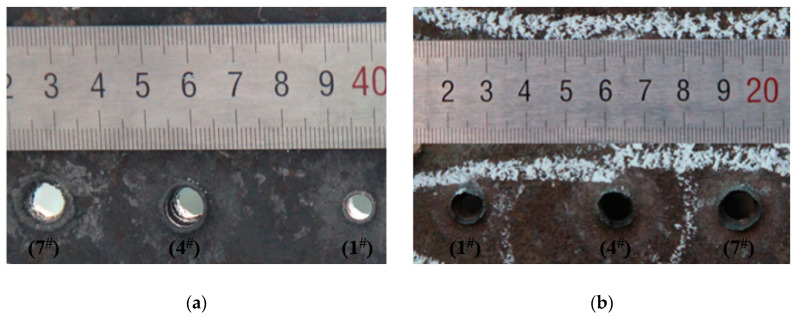
Damage caused to the RHA target plate: (**a**) Front; (**b**) Back.

**Figure 7 materials-14-00721-f007:**
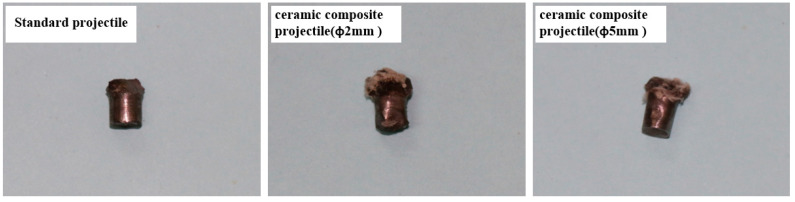
Recovered projectile cores after penetration of the armored steel target.

**Figure 8 materials-14-00721-f008:**
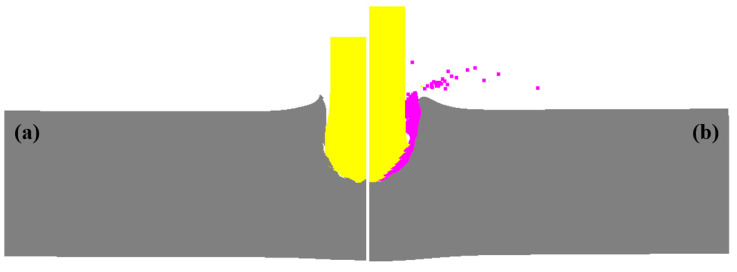
State comparison diagram of two projectiles penetrating to the half-depth of an armored steel target: (**a**) Standard projectile; (**b**) Ceramic composite projectile (Ø5 mm).

**Figure 9 materials-14-00721-f009:**
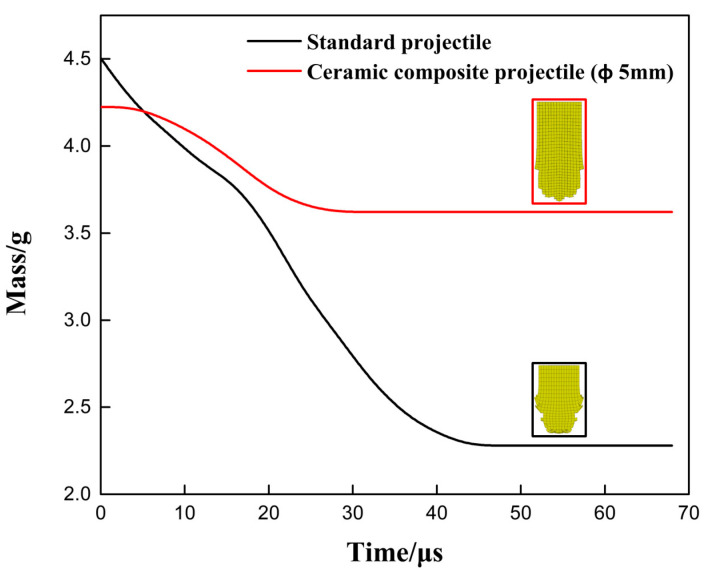
Projectile Core Mass-Time Curve.

**Figure 10 materials-14-00721-f010:**
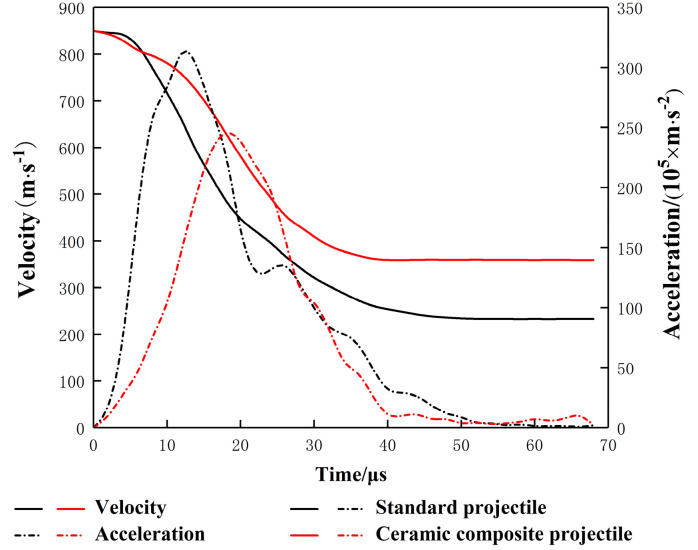
Projectile Core Velocity/Acceleration–Time Curve.

**Figure 11 materials-14-00721-f011:**
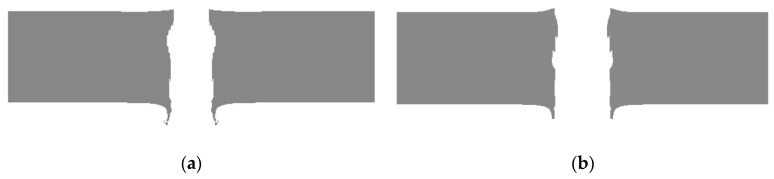
Damage effects of projectiles on RHA target plates: (**a**) Standard projectile; (**b**) ceramic composite projectile (Ø5 mm).

**Figure 12 materials-14-00721-f012:**
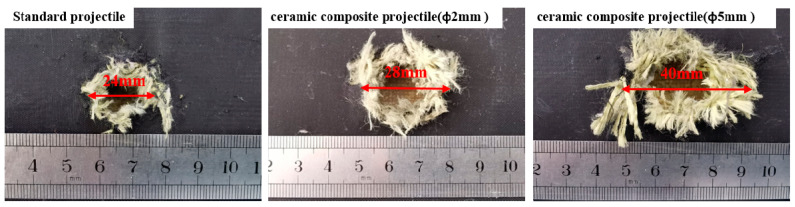
Damage caused to the Ceramic/Kevlar composite armor surface.

**Figure 13 materials-14-00721-f013:**
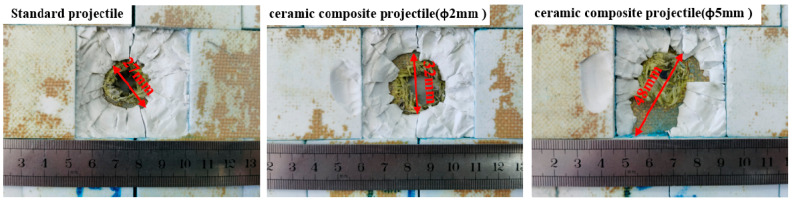
Ceramic/Kevlar composite armor target fragments.

**Figure 14 materials-14-00721-f014:**
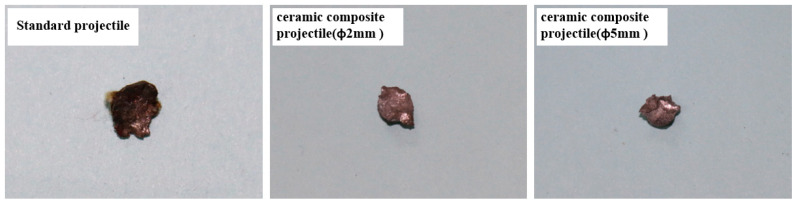
Projectile core residual after the penetration of the Ceramic/Kevlar composite armor.

**Figure 15 materials-14-00721-f015:**
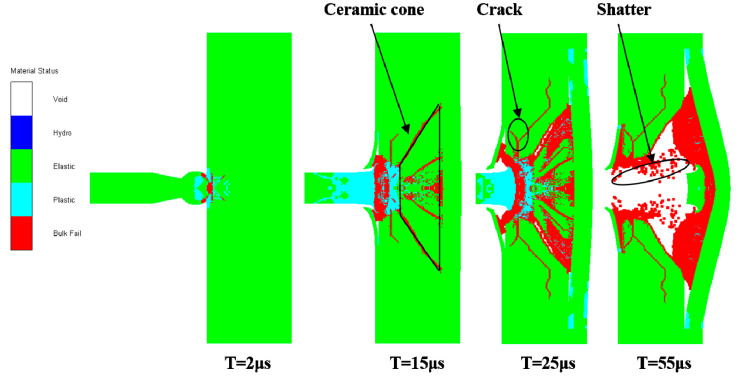
Penetration Process of the Ceramic Composite Projectile into the Ceramic/Kevlar Composite Armor.

**Figure 16 materials-14-00721-f016:**
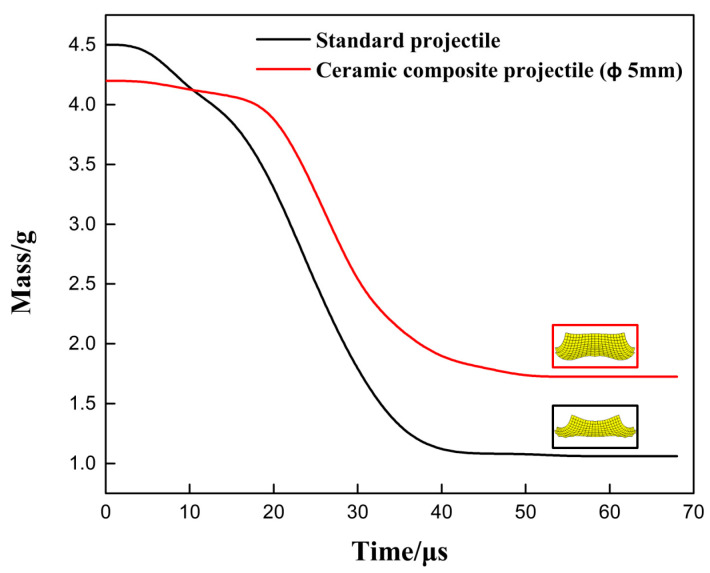
Projectile Core Mass–Time Curve.

**Figure 17 materials-14-00721-f017:**
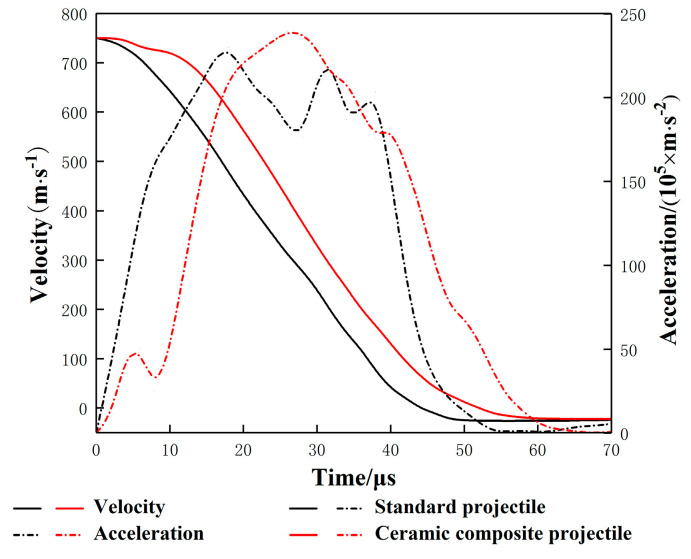
Projectile Core Velocity/Acceleration–Time Curve.

**Figure 18 materials-14-00721-f018:**
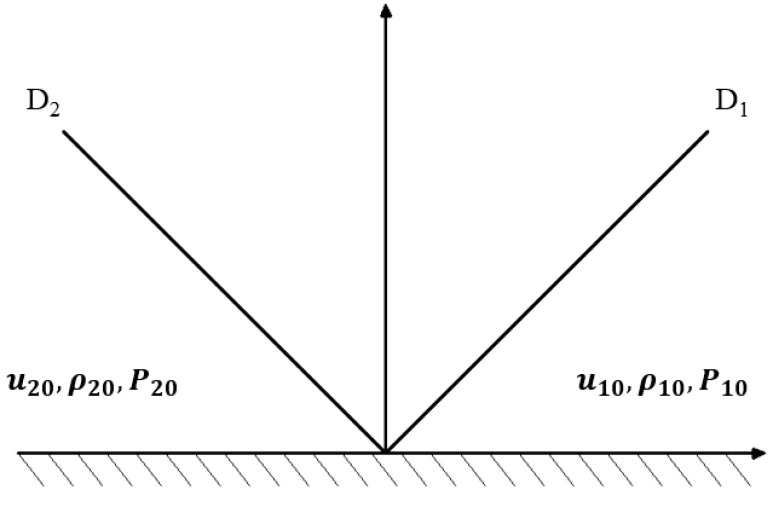
The left and right characteristic lines of the projectile for the collision of different media.

**Table 1 materials-14-00721-t001:** Parameters of the ZrO2 ceramic material.

Materials	ρ/(g·cm^−3^)	G/Armor	HEL/Armor	A	B	C	M	N
ZrO_2_ Ceramic	5.90	152	6.57	0.93	0.72	0.007	0.38	0.64

**Table 2 materials-14-00721-t002:** Test data of projectiles with different structures penetrating the 10 mm armored steel target.

Projectile Structure	Number	Projectile Mass/g	ZrO_2_ Ceramic Mass/g	Tungsten Core Mass/g	Initial Speed/(m/s)	Residual Core Mass/g	Damage of Armored Steel Target		
							Hole Entrance/mm	Hole Entrance Flange/mm	Hole Exit/mm
Standard projectile	1	4.5	-	4.5	842	2.29	Ø5.03	Ø7.21	Ø7.53
	2	4.5	-	4.5	851	2.08	Ø5.16	Ø7.45	Ø7.68
	3	4.5	-	4.5	846	2.21	Ø5.06	Ø7.19	Ø7.61
ceramic composite projectile (Ø2 mm)	4	4.5	0.03	4.47	848	3.26	Ø7.52	11.3 × 12	8.98 × 7.96
	5	4.5	0.03	4.47	842	3.19	Ø7.36	10.6 × 11.2	8.68 × 8.16
	6	4.5	0.03	4.47	854	3.07	Ø7.43	11.5 × 11.7	9.08 × 8.06
ceramic composite projectile (Ø5 mm)	7	4.5	0.38	4.12	852	3.64	Ø8.11	Ø14.01	10.35 × 9.81
	8	4.5	0.38	4.12	843	3.66	Ø7.86	Ø13.92	10.13 × 10.01
	9	4.5	0.38	4.12	844	3.52	Ø7.93	Ø13.68	10.26 × 9.97

**Table 3 materials-14-00721-t003:** Test and simulation data of projectiles with different structures penetrating a 10 mm RHA.

Projectile Structure	Simulation			Test		
	Hole Entrance/mm	Hole Exit/mm	Residual Core Mass/g	Hole Entrance/mm	Hole Exit/mm	Residual Core Mass/g
Standard projectile	Ø5.09	Ø7.59	2.28	Ø5.06	Ø7.19	2.21
ceramic composite projectile (Ø5 mm)	Ø8.07	Ø10.01	3.62	Ø7.93	10.26×9.97	3.52

**Table 4 materials-14-00721-t004:** Test results of projectile cores penetrating the ceramic/Kevlar composite armor in different structures.

Projectile Structure	Number	Full-Projectile Mass/g	ZrO_2_ Ceramic Mass/g	Tungsten Core Mass/g	Initial Speed/(m/s)	Residual Core Mass/g	Damage Effect
Standard projectile	1	4.5	-	4.5	754	1.07	Ceramic/Kevlar composite armor ceramic plate broken into large fragments. Kevlar bulges on the back.
	2	4.5	-	4.5	752	1.12	
	3	4.5	-	4.5	746	1.09	
Ceramic composite projectile (Ø2 mm)	4	4.5	0.03	4.47	750	1.23	Ceramic/Kevlar composite armor ceramic plate broken into small fragments. Kevlar bulges on the back.
	5	4.5	0.03	4.47	744	1.32	
	6	4.5	0.03	4.47	747	1.26	
Ceramic composite projectile (Ø5 mm)	7	4.5	0.38	4.12	751	1.67 g	Ceramic/Kevlar composite armor ceramic plate broken into small fragments. Kevlar bulges on the back.
	8	4.5	0.38	4.12	753	1.62	
	9	4.5	0.38	4.12	746	1.59	

**Table 5 materials-14-00721-t005:** Hugoniot Parameters of Materials.

Name	RHA	Ceramic Target	Ceramic Composite Projectile	Standard Projectile
Density ρ/(kg/m^3^)	7850	3630	16,000	17,600
Material parameter α/(km/s)	3.57	5.65	7.68	5.124
Material parameter b	1.92	1.65	1.65	1.233

## Data Availability

The data that support the findings of this study are available from the corresponding author (K.R.), upon reasonable request.

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
