# Peer review of "Study on the Penetration Performance of a 5.8 mm Ceramic Composite Projectile"

_materials, 2021, doi:10.3390/ma14040721_

Round 1

Reviewer 1 Report

The article concerns the identification of the process of penetration of the selected ballistic shield by the bullet. It is a very complex process and its analysis is very difficult. In the introduction, the authors conducted very limited studies on the current state of knowledge. In the main part the authors presented the effects of experimental research on the penetration of various ballistic shields with different missiles using standard methods. They also presented limited simulation results without describing the simulation model itself. The presented test of mathematical description of impact pressure is not convincing and requires important additions and explanations. Moreover, the authors have not shown what the results from these considerations are needed for. The conclusions presented in the summary are superficial and do not fully present the significance of the issues raised in the article. Full understanding of the article also requires explanations to the following doubts:

- line 49 – how to understand, that the appearance of standard projectiles with a ZrO2 were not changed?

- line 72 – what standards meets a ceramic ball in the tests?

- line 79-80 – what tests were done to gather data concerning relative density, bending strength and breaking tenacity?

- fig. 1b – what is a scale factor in the picture? Areas mentioned in the text and pointed in the picture would be useful,

- fig. 2 – there are a various scale factor in the pictures – a) 1 μm, b) 20 μm – that makes analysis difficult,

- fig. 1a and fig. 2a – what is the reason to present pictures of the some material twice?

- line 93 – if the total mass of projectile is the same when a ceramics was added, are there projectiles really according to the standard? What is the mass of metal and ceramic in each case?

- line 97 – what kind of 10 mm steel was use in the tests? What was the layout of Kevlar fibers in each layers? What kind of ceramic were used in the tests? Material data should be presented.

- line 100/fig. 4 – how to understand that 12,7 mm ballistic gun was to 5,8 mm projectiles?

- pic. 4 (Projectile) – there are some others projectiles presented in the pictures which are not mentioned in the text.

- line 108 – the information about FEM model are necessary to present. How the contact between projectile and the armor was set up? Why ø5mm projectile was used in simulation? Previously 5,8mm was described.

- line 111-113 – there are very little important information about the condition of the simulation – some explanation about SPH method used in the test would be necessary as well as about the pressure outflow exerted from the model

- table 2 – what is the reason to put the residual core mass in the table, if there is no information about initial mass of cores? How the hole sizes were measured with so high precision? Methodology would be useful. How to explain, that the size of entrance flanges are bigger than exit holes for projectiles with ceramic? What was the residual speed of projectiles?  Necessary to compare with pic. 10 and 17.

- pic. 6 – there are no comments in the text. So, what was the reason to put pictures in the paper?

- line 147 – The sentence: This is because, … - if there is a statement – any proof is necessary or if there is a guess – question mark at the end of sentence is necessary.

- pic. 8 – please explain the fact, that the height of metal part of ceramic projectile is significantly higher than metal one? Pic. 3 and 7 presents a different dimensions.

- line 167 – how to understand: (…) accelerated velocity – time curve

- pic. 11 – Do the pictures present the FEM model of damage or experimental one? – explanation is necessary. What are the dimension along the penetration? Are the standard projectile hole dimension comparable with data presented in table 2?

- line 214-216 – What does it mean, that: “The damage effect of ceramic …” – the criteria were taken into considerations?

- line 216-218 – “By comparing the damage of …” – explanation is needed, what was taken into considerations to got the point.

- line 219 – “Figure 12 shows the damage …” – there is no visual differences between holes in the pictures in pic. 12, the clues are not proofed with use the pic. 12. Explanation is needed.

- line 220 – “According to the ceramic fragments in Fig. 13 …” - - there are no visual fragments in pic.13, the clues are not proofed with use the pic. 13. Explanation is needed.

- line 223 – How to assess the mass using pictures presented in pic. 14?  Explanation is needed.

- line 234 – What was the reason to conduct simulation with different speed (750 m/s) then in the experiment (850 m/s)?

- pic. 15 – there is a mistake – time of penetration – t=2μs, and in the text (line 235) t=5μs.

- pic. 17 – the initial acceleration of standard projectile is 100 m/s2, the final velocity is higher than zero and is getting higher from t=57μs (ceramic projectile) – projectiles penetrated the armor? (pic.  13 presents, that no) - Explanation is needed.

- line 263 – How to understand the statement – the ceramic head protects the metal core to a certain extent. - Explanation is needed.

- line 263 – How to understand the statement – “Therefore, the structure …” – authors mentioned formerly, that ceramic projectile penetrates the armor better than standard one. Explanation is needed.

- line 265 – 308 – There is a lack of explanation, that Hugoniont states are valid in this kind of phenomenon, like penetrating the armor by ceramic projectiles. This part of paper need to be extended and better link to references.

Reviewer 2 Report

Introduction: 27/28 remove 'shock' or replace by 'impact'

29: use 'erosion' instead of 'deformation' (AP cores do not deform)

32: a ceramic component, ....

33: ceramics ate not known for their low cost....(compared with steel)

all over in paper: symbol for diameter is not phi φ but rather Ø

DOP should be explained

2.1: 54-57 provide reference or leave out sintering Temp info.

64/65: 3% Al2O3 is added to ZrO2, not opposite as stated in paper.

80: explain 'breaking tenacity' (or do you mean fracture toughness Kic)

87: remove 'granular' and provide reference for statement on energy consumption

97: were applied as targets (remove 'for the testing') 

3.1: 108/109: ...their penetration ability against RHA ....

Figure 7: Recovered projctile cores after penetration of armoured steel target

4.2.1: 177: ....seen that acceleration of the ....

Figure 12: Damage of Ceramic/Kevlar....

260: experimental and simulation results

269: The shock wave velocities....

5.2: 283: Shock wave and particle velocity

287: do not (NOT don't)   293: shock wave velocity

297: Hugoniot relation of shock loaded matter.   ..shock wave pressure..

302: interfacial pressure is equal to the velocity ??? linear dependent ??

307: shock wave

5.3: 310: projectile impacted RHA

Table 5: ceramic composite projectile density is 5900 (NOT 16000)

314: Lingo ???  Provide Pimpact, not the difference between two conditions

Reviewer 3 Report

Briefly, this paper focuses on the high-velocity impact analysis of ceramic balls into armor steel and ceramic/Kevlar composite. The ceramic composite projectile has been studied via (DOP) experimental and numerical simulation.

Comments:

  1. First of all, English is not up to the standard of the journal. This is a serious blocking point that requires an intensive reworking action from the authors with the help of a native speaker.

  1. There are some relevant experimental and numerical results that can be combined into a table or graph.

Based on that, I strongly suggest providing extra validations to demonstrates the difference between those approaches.

Reviewer 4 Report

1.The reviewer believes that in the Introduction, the authors should point out the disadvantages of using tungsten and depleted uranium as cores for armor-piercing projectiles, which forces researchers to look for a replacement for them.

The reviewer believes that the authors of the article should explain to the readers what limitations of the used cores they forced to look for a replacement for them.

Reviewer 5 Report

This paper aims to improve the penetration ability of 5.8mm standard projectile by inserting a ZrO2 ceramic ball with high hardness, high temperature, and pressure resistance at its head. The study tried to analyses the DOP of the enhanced projectile (ceramic composite projectile) and the standard through experiment and numerical simulation by tested on the Rolled Homogeneous Armor (RHA) and ceramic/ Kevlar composite armour. In general, the paper raised some interesting points but it also needs some work to address the intended information for the audiences.

My Comments and suggestions are the followings

 General

 The author might be very focused on writing the information on the manuscript. However, please use concise and formal English to make the readers understand in some paragraph. Moreover, the authors are also advised once more to take some time to read and correct some grammatical errors.

Specific Comment

Abstract

  1. The abstract needs a little bit more information on the methods used and numerical values of the result in a concise and clear manner.

Introduction

  1. The author provided a very limited state of the art of the topic. It needs a little more extended literature on the topic.
  2. Please also bring some of the sentences of section 2.1 (especially paragraph one) to the introduction part.

Experiment preparation

  1. Did you produce the ZrO2 ceramic ball by yourself? If yes, could you please put some figures which backed your procedures of manufacturing?
  2. What refers to the ceramic body in Page 2 line 72?
  3. What makes the referred projectile core as standard? Any reference? please mention. Page 3 Line 95.
  4. Could you give more information (composition, material, density etc.) regarding the 10mm rolled homogeneous armour? Page 3 Line 100.
  5. You also mentioned ‘Ceramic/ Kevlar composite armour has two layers of Kevlar on its front, Al2O3 ceramic with 10mm thick and 50mm×50mm side length in the middle, and 10 layers of Kevlar on its back’. Page 3 Line 100-104. My suggestion is that there are various Kevlar fibres with different parameter. Besides, if you use Kevlar fabrics, which fabric type, structure, density……. . In general, please clear the material (specifically the Kevlar) properties.
  6. Figure 4 is not clear and need more clarification. Page 3.

Results and Analysis

  1. What ‘Number’ refers in Table 2, column 2? Are they the different structure of projectiles or shot number? Please clarify.
  2. You have measured the values of damages of armoured steel target using an ordinary ruler (Table 2). How precise is to use and measure such small and similar damages with such measurement system?
  3. Figure 6. Fracture Morphology Images. Page 5, Line 144. But it is not fractured morphology, it is penetrated holes.
  4. You mentioned ‘after the aperture of the target plate is measured, it is found that the average penetration area of ceramic composite projectiles (Ø5mm & Ø2mm) is 2.46 times and 2.14 times that of the standard one, respectively’ Page 5, Line 153 -155. How do you measure the area? What does it mean the average here (As far as the information, each projectile was one-shot)?
  5. You also mentioned ‘As shown in Figure 7, the average residual mass of projectile cores recycled from ceramic composite projectile …..’. Page 5 Line 155 – 157. First, how did you get the average residual mass for tested projectile since the shot was one for each? Second, the figure does not show the average mass of each projectile rather shows the images of different projectile residuals.
  6. What is the reason behind the residual mass of the ceramic composite projectile core is remarkably larger than that of the standard one? Page 6, Line 158 – 159.
  7. What is the reason behind the residual length of the ceramic composite projectile (Ø5mm) core is prominently longer than that of the standard one? Page 6, Line 165 - 166.
  8. Please re-write the paragraph. Page 7, Line 198-200.
  9. Which standard projectile (1, 2 or 3) effects are discussed and indicated in Figure 12, 13 & 14?
  10. Do you think the measured damage diameter is precise and relevant Figure 12? It is also possible only showing the damages without measurement values (since you are trying to show the severity of the damage zone). But it will be interesting if your analysis the failure mechanisms of the target with close looking by Optical Microscope or SEM.
  11. ‘As shown in Figure 14, the average residual mass of recycled project…..’ Page 6, Line 165 – 166. But, as I mentioned above, there are no average values in your test.

Round 2

Reviewer 1 Report

After reading the answers to the questions posed previously, many additional doubts arise as to the correctness of the research and analysis of the results obtained. I suggest the authors to try again to thoroughly analyze the research and describe the results in a clear way. Maybe it is worth considering publishing the article in a less demanding journal.
